# Usage patterns of aromatherapy essential oil among Chinese consumers

**Jun Xiao, Satoshi Nakai** [ID]*

Graduate School of Environment and Information Sciences, Yokohama National University, Kanagawa, Japan

* nakai-satoshi-dc@ynu.ac.jp

## Abstract

Given the concern over contact allergy risk associated with aromatherapy, information regarding the use of essential oils (EOs) is crucial for consumer dermal exposure assessment. In this study we mainly aim to describe the usage patterns of EOs among Chinese consumers to provide important data for exposure assessment to fragrance allergens in EOs. A web survey was conducted in April 2020 among 1,518 potential Chinese EO consumers to assess consumer usage patterns. The usage patterns of 11 types of EOs were collected among female consumers (N = 457; ages 0–70). For females aged 0–14, they used Lavanda (42.9%) and Tea tree (57.1%) oils only. Among the senior age groups (15–70), Lavanda oil was the most used EO with 46.7%, 51%, 68.1%, and 50% for females aged 15–24, 25–39, 40–59 and 60–70, respectively. The majority of females aged 25–59 used Rose, Lavanda, Sandalwood, Frankincense and Jasmine oil on their whole face more than three times a week at diverse dilution rates. Usage patterns are described for all age groups. In consideration of usage pattern of females aged 25–59, co-exposure to fragrance allergens contained in EOs and cosmetics could make them vulnerable to contact allergy. This study provides valuable information for dermal exposure assessment.

## Introduction

Aromatherapy is the use of essential oils (EO) extracted from flowers, bark, stems, leaves, roots or other parts of a plant to balance, harmonize and offer psychological and physical benefits through oral intake, massage, topical application, or inhalation [1, 2].

Research from a domestic personal care manufacturer AFU in China stated that the sales of the aromatherapy products increase from 0.0743 million to 19.318 million USD between 2010 and 2017 [3]. The sales data of EOs for households show that EO consumption is relatively rampant in China, particularly among females.

However, EOs are composed of naturally presenting fragrance allergenic compounds, such as limonene, citral or oxidized linalool [4, 5]. Although such allergens may have the potential to cause sensitization, they can be formulated into cosmetics or detergents at safe levels by complying with specific regulations [5]. However, there are no specific regulations to limit contact allergens in aromatherapy EOs. Consequently, skin sensitization may result after

---

**Data Availability Statement:** All relevant data are within the paper and its Supporting Information files.

**Funding:** The authors received no specific funding for this work.

**Competing interests:** The authors have declared that no competing interests exist

dermal exposure to EOs when EOs are inappropriately used. Skin allergy due to tropical use of EOs has been reported in publications [6–9].

Skin sensitization has a significant impact from the public health perspective, either in terms of financial impact or the impaired quality of life of patients [5]. From a quantitative perspective, EO dermal exposure has been demonstrated as a key risk factor in the induction of skin sensitization [10, 11]. Therefore, this exposure should be determined to assess the risk and better protect the consumer. Information on the usage patterns of EOs is necessary to assess the corresponding consumer exposure.

In recent years, studies have investigated the usage patterns of many cosmetic products among various populations in different regions, providing valuable information for exposure and risk assessment for substance contained in cosmetics [12–15]. But for aromatherapy products, studies conducted on Australian pregnant women [16] and Japanese university students [17] gave data on the prevalence of aromatherapy among restricted sub-population. Until 2016, a survey of usage patterns of aromatherapy conducted among the French general population [18] provided important information regarding dermal exposure to fragrance allergenic molecules in EOs. However, there is a lack of published usage-pattern/exposure data 1) on Chinese consumers whose consumption of EOs may be different from that of French consumers and 2) on specific application body areas according to each type of EO. Most importantly, because consumers buy EOs and dilute them in bases (e.g., vegetable oils or skin care products) at varied dilution rates, data on dilution rates of use should be more accurate for calculating the amount of use (i.e., the number of drops of EO). Thus, the dilution rates of use rather than the amount of bases should be included in the survey on usage patterns of aromatherapy. The aim of our study was to describe the usage patterns of EOs among Chinese consumers to provide important information on percentage of users with regard to types of EOs, exposed body areas, and dilution rates of use, as well as frequency of use per EO for assessing dermal exposure to fragrance allergens.

## Materials and methods

### Study population

Chinese Aromatic & Aromatherapy WeChat groups are chatting groups established by the Chinese Aromatic & Aromatherapy Association. In this platform, there are approximately 1,500 EO consumers registered to share information on EOs, including safety precautions in aromatherapy practice, the receipt of EOs, the methods of EO storage, and the quality problems of EOs. However, the sociodemographic data on these consumers is unavailable. Thus, approximately 1,500 consumers registered in Aromatic & Aromatherapy WeChat groups were invited to participate in our survey. Considering that EOs are used independent of age, for children (0–14) using EOs daily, but not registered in WeChat groups, their registering adults were asked an agreement to complete the survey on behalf of the children. Thus, the study population consisted of several age groups, which allowed us to examine the variability of use patterns across age.

### Data collection

In late March 2020, all registered EO consumers in Chinese Aromatic & Aromatherapy WeChat groups were informed of the overall objectives of this survey. In April 2020, upon releasing the direct link to the website hosting the questionnaire, an invitation was sent to everyone. To ensure the response rate, the links and invitations were sent to WeChat groups once a day and every day during the two-week period of the survey.

We developed a web-based questionnaire to determine the use patterns of EOs, specifically through the dermal route. This questionnaire contained general questions regarding demographics, body characteristics, and EO consumption data. The detailed usage patterns of EOs were assessed using questions concerning the types of EOs, body areas of application, frequency of use, and dilution rates of use for each EO. With respect to the types of EOs, 11 types of EOs (Rose, Lavanda, Tea tree, Ginger, Mentha, Lemon, Sandalwood, Frankincense, Ylang ylang, Eucalyptus, and Jasmine) were given as choices. These EOs were selected because they were the most popular EO among the panel of respondents in our pre-research survey. Skin sensitization has been shown to be caused by the migration of allergens to the local lymph nodes where the product and allergens were applied [11]. Therefore, the body areas the EOs were applied to should be determined to calculate the dermal exposure to fragrance allergens. Thus, we include different parts of the body as multiple choices to determine the body areas that each EO will be applied. Given that EOs are suggested to be diluted in bases, we developed questions and multiple choices to describe the dilution rates in bases, that is, less than 1% (less than 1 drop in 5 mL base oil), 1% (1 drop in 5 mL base oil), 2% (2 drops in 5 mL base oil), 3% (3 drops in 5 mL base oil), 4% (4 drops in 5 mL base oil), 5% (5 drops in 5 mL base oil), more than 5% (more than 5 drops in 5 mL base oil), adding EO to cosmetic products, and undiluted (respondent was asked to give the number of drops in the latter two choices). The amount of bases per use was not included in our survey, consequently, these data were obtained from additional sources.

Such multiple choices were determined by closely observing safety precautions and recommendations provided on aromatherapy websites and widely consulting professional aromatherapists. Additional detailed information about the questionnaire is provided in S1 Appendix. The study was approved by the Ethics Committee of Yokohama National University (No. non-medical-2019-17) in February 2020. All methods were performed in accordance with relevant guidelines and regulations. All subjects involved in the study have been properly instructed and they have indicated that they give their consent for information about themselves. The adults have been asked an agreement to give their consent on behalf of their children.

## Data treatment

To summarize the demographic characteristics of the total consumers, basic descriptive statistics were used. With respect to the geographical data, the results of the living city were divided into four economic regions based on per capita gross regional product and indices by the National Bureau of Statistics of the People's Republic of China in 2013 [19]: east, central, northeast, and west.

The same data analysis was conducted for the percentage of dermal users sorted by sex and age groups (0–14, 15–24, 25–39, 40–59, and 60–70).

To provide a better overview of the use patterns of EOs among Chinese consumers, we chose to describe the proportions of respondents on the type of EOs, exposed body areas, dilution rates of use across age groups, and frequency of use for each EO. Data on the type of EOs, exposed body areas, dilution rates of use are treated as non-parametric. And data on frequency of use for each EO is treated as parametric. The chi-squared or Fisher's test was conducted to compare the differences in percentages of users on the type of EOs, exposed body area, and dilution rates of use across age groups.

To calculate the frequency for each EO used, responses were replaced by frequencies per day (daily replaced by 1, weekly by 1/7, monthly by 1/30). The response "yearly" was considered null and replaced by 0. Some of the respondents provided the number of times the EO was used; then, the frequency was multiplied by the value of times. The values for the number of

times were assigned using the following criteria: For answers such as "X-Y times," the value was replaced by the average value; that is, (X + Y)/2. Then, for answers such as "more than Z," the value was replaced by Z. The Kruskall-Wallis test was conducted to compare difference in frequencies of use per age and per type of EO among females. When no statistical differences were observed, the sample was pooled per type of EO. The pooled samples of responses were used to calculate the mean, standard deviations (SD), 25th percentile (P25), 50th percentile (P50), 75th percentile (P75), 95th percentile (P95), and 99th percentile (P99) values per type of EO.

Statistical data analyses were performed using IBM SPSS statistics 26.0.0.0 (IBM, Armonk, USA). Differences with a *p*-value of less than 0.05 were considered significant.

## Results

### General demographics

In total, 534 out of 1,518 potential participants (after exclusion of 44 invalid respondents) completed the web survey, which correspondents to a response rate of 35.2%. The sex distribution was compared among age (0–14, 15–24, 25–39, 40–59, 60–70), pregnancy status, profession, and economic region (northeast, center, east, and west) groups. In our population sample, female consumers were overrepresented (N = 481; 90.1%). In addition, the age groups (0–14, 15–24, 25–39, 40–59, 60–70) represented 15.09%, 9.43%, 30.19%, 39.62%, and 5.66% of the male consumers, respectively, and the age groups for females were 2.08%, 4.57%, 63.41%, 28.69%, and 1.25%, respectively. The proportion of salaried males and females were the highest (33.96% and 30.77%, respectively) in socio-professional status. The majority of the consumers were from the east of China for male (67.92%) and female (87.94%) consumers (S1 Table).

### Prevalence of dermal use

The prevalence of dermal use was defined as the percentage of users exposed through the dermal route calculated based on the total number of consumers. Table 1 shows the percentages for male and female dermal users related to the total number of consumers per age group. Generally, 70% of male respondents (N = 37 on 53) and 95% of female respondents (N = 457 on 481) were exposed through dermal route.

### Usage patterns among female dermal users

As the female dermal users were overrepresented (N = 457; 92.5% of total dermal users), usage patterns were described on female dermal users only. The percentages of females on the types of EOs, exposed body areas, and dilution rates were calculated based on 457 female dermal users.

**Types of EOs applied on the skin among females.** Fig 1 shows the percentage of female dermal users by age groups for 11 types of EOs. The sum of percentages is over 100% because of

**Table 1. Percentage of users through dermal route per age group.**

| Age groups | Males (N = 53) | | Females (N = 481) | |
|---|---|---|---|---|
| | Number of users | Frequency of dermal users N(%) | Number of users | Frequency of dermal users N(%) |
| 0–14 years | 8 | 7 (87.5%) | 10 | 7 (70%) |
| 15–24 years | 5 | 4 (80.0%) | 22 | 15 (68.1%) |
| 25–39 years | 16 | 9 (56.2%) | 305 | 294 (96.3%) |
| 40–59 years | 21 | 16 (76.1%) | 138 | 135 (97.8%) |
| 60–70 years | 3 | 1 (33.3%) | 6 | 6 (100%) |

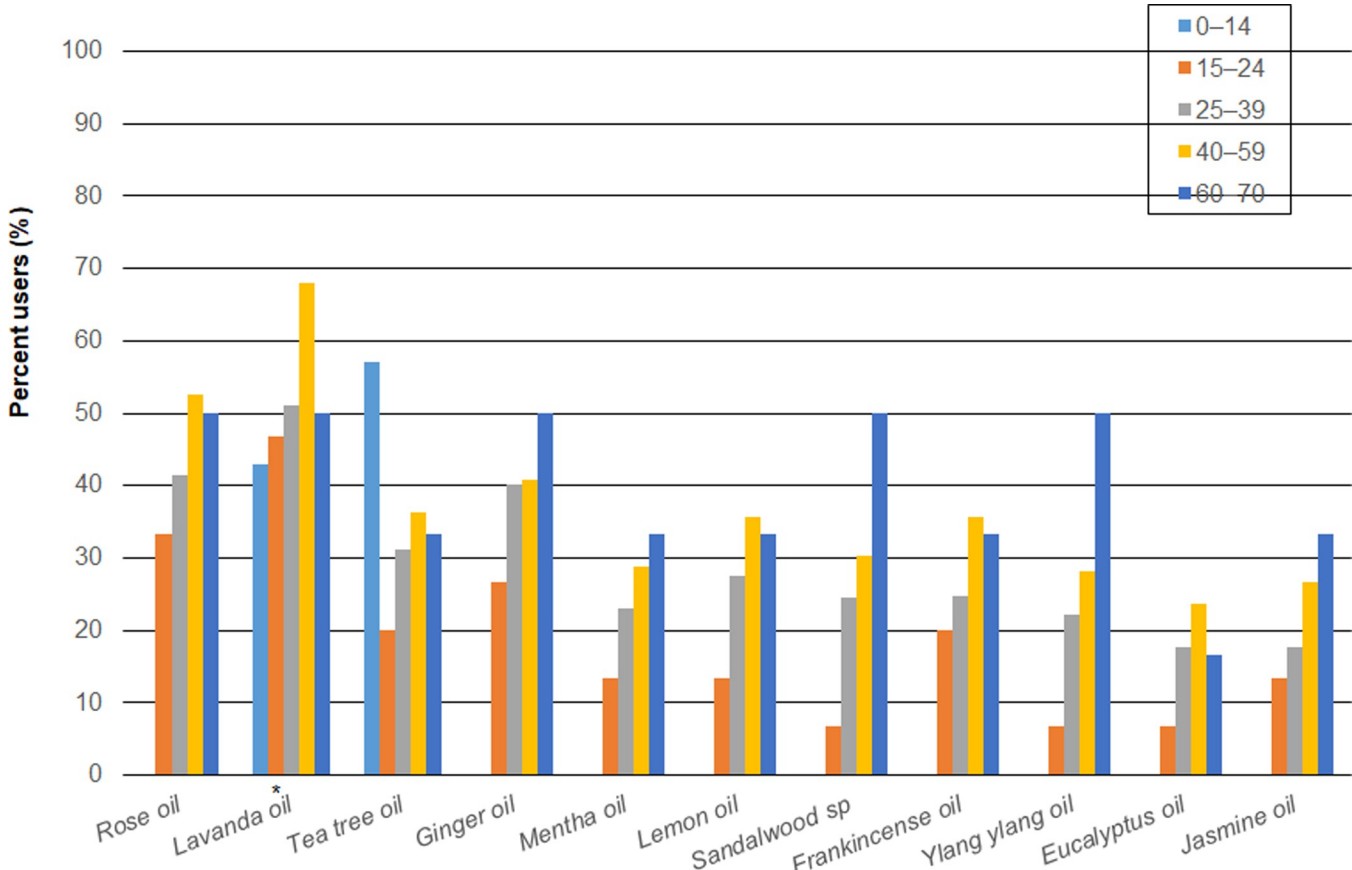

**Fig 1. Percentage of female dermal users by age groups for 11 types of essential oils.** Significant statistical differences (p < 0.05) marked with *.

the multiple answers allowed per consumer. For females aged 0–14, they used only two types of EOs: Lavanda (42.9%) and Tea tree (57.1%) oils. Among the senior age groups (15–70), Lavanda oil was the most used EO with 46.7%, 51%, 68.1%, and 50% for females aged 15–24, 25–39, 40–59 and 60–70, respectively. Then, Rose oil was used by 33.3% (15–24), 41.5% (25–39), 52.6% (40–59), and 50% (60–70) of female users. The percentage of females aged 40–59 who used Lavanda oil was higher compared with other age groups (p < 0.05). However, for the other 10 types of EOs, no significant difference was observed among different age groups. Although for some EOs (i.e., Tea tree and Sandalwood oil), the percentages of females aged 0–14 or 60–70 were evidently higher than other age groups because of small sample size of such groups (only seven aged 0–14 and six aged 60–70), no significance was observed.

**Exposd body areas.** Fig 2 shows a heat map providing the percentage of users who applied 11 different EOs on 15 body parts classified by age groups. According to our survey, female consumers aged 0–14 used Lavanda and Tea tree oils only, and 43% of them applied these two EOs on the breast/chest and back. Consumers aged 25–39 and 40–59 used EOs on almost all listed parts of the body. However, the majority of them applied Rose, Lavanda, Sandalwood, Frankincense, and Jasmine on the whole face. Compared with the diversities in body areas exposed by such age groups, the body areas of application of females aged 15–24, and 60–70 were relatively restricted. For example, women aged 60–70 tended to apply EOs on their face and lower limbs (thighs, calves/shins, and feet). Except for the philtrum, wrists, breast/chest, and back, significant differences were found among different age groups when they used

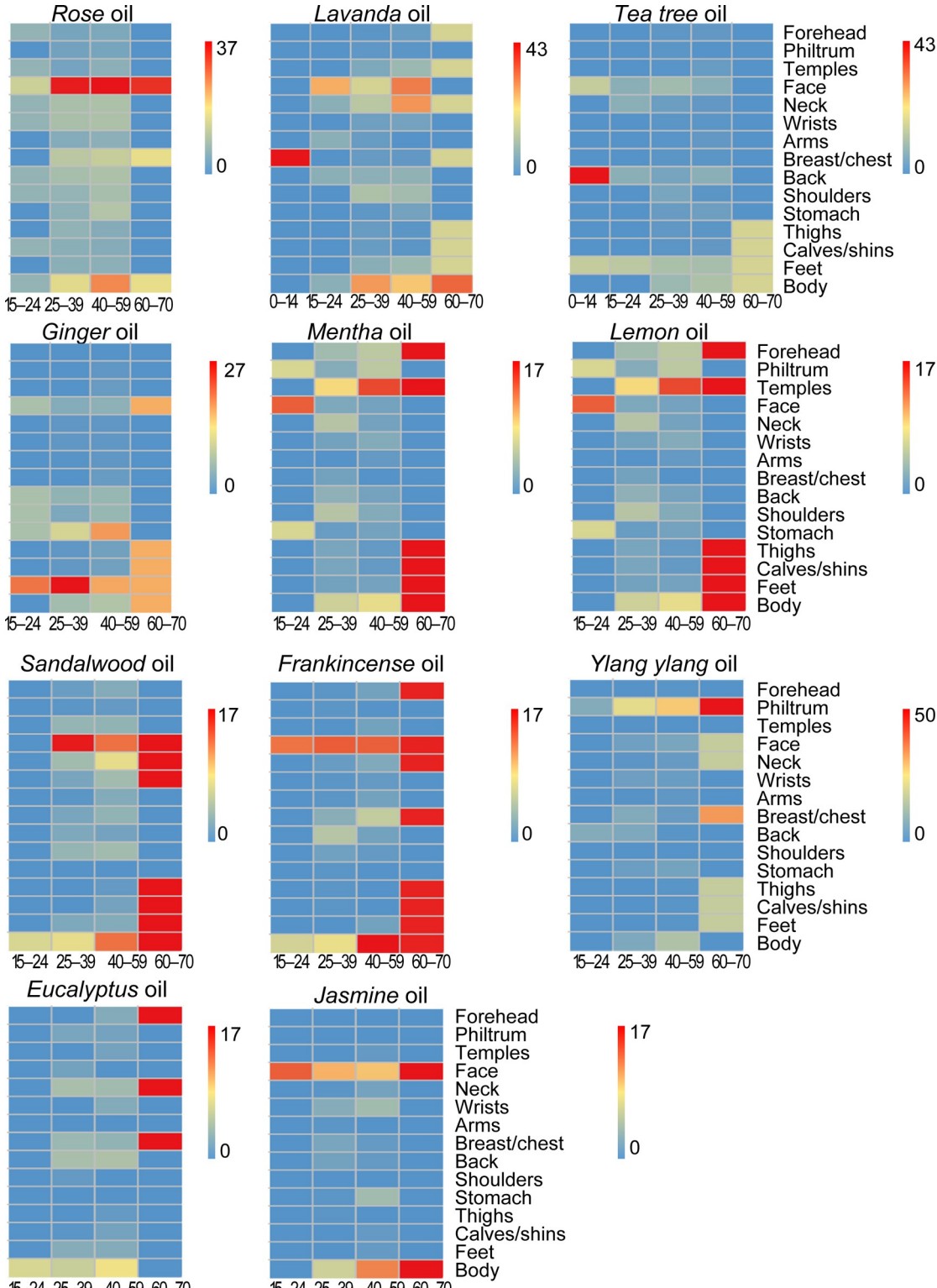

**Fig 2. A heat map is used to show the percentage of users who used each essential oil (EO) on different parts of the body (y-axis) from different age groups (x-axis).** The redder cells indicate that the higher (highest) percentages of users applied EOs on parts of the body across age groups. For example, the majority of females (27%) aged 25–39 applied Ginger oil on their feet.

Lavanda oil. Significant differences were also observed when they used Rose oil on the face, Ginger oil on the feet, and Sandalwood oil on the face (p < 0.05).

The following should be noted:

1. Females aged 0–14 used Lavanda and Tea tree oils only;

2. Scales are not unified. Ununified scales gave the reddest cells per each EO, which the extreme values on the most exposed body areas for each EO across age groups were easily found.

**Frequencies of use.** Table 2 shows the mean, SD, and selected percentile (P25–P99) values for the frequency of use for each type of EO. N is the number of female dermal users per EO. All values are represented on day$^{-1}$. For example, on average, female users applied Lavanda 0.68 times per day, with a P50 and P95 twice a week (0.28) and twice a day (2.00), respectively. Rose oil was used at a mean frequency of 1.06 times per day as the highest frequency. Lavanda, Tea tree, Sandalwood, Frankincense, and Eucalyptus were the EOs with similar mean values of 0.57–0.68 representing four to five times a week. Moreover, according to the mean value of the use frequency, females used Mentha, Lemon, Ylang ylang, and Jasmine less frequently compared with other EOs. Notably, that for Eucalyptus and Jasmine oil P99 values are not given.

**Dilution rates of use.** In this study, we provided the distribution of the prevalence of use and dilution rates of EOs in bases for all 11 types of EOs per age group in Fig 3.

The percentage (42.9%) of Lavanda oil users at <1% dilution rates was higher in aged 0–14 compared with that of the senior age groups (p < 0.05). However, no significant difference was observed in the percentages of users for the other 10 EOs across age groups. Then, 42.9% of children (0–14) used <1% dilution of Tea tree oil. Moreover, for the senior age groups (25–59), generally, the dilution rates differed within the types of EOs, especially for Rose, Lavanda, Lemon, Sandalwood, Frankincense, Ylang ylang, and Jasmine oil. Such EOs were also frequently applied on the whole face or parts of face. For example, as for Lavanda oil, 6.8%, 5.8%, 16.3%, 3.1%, 1%, 7.8%, 2.7%, 2.4%, and 5.1% of females aged 15–24 used at <1%, 1%, 2%, 3%, 4%, 5%, and >5% dilution; adding EO in cosmetic products; and undiluted, respectively. From another perspective, for the other EOs, a higher percentage of consumers aged 15–59

**Table 2. Frequency of use (per day) for each essential oil (EO).**

| Types of EO | Mean | SD | P25 | P50 | P75 | P90 | P95 | P99 |
|---|---|---|---|---|---|---|---|---|
| *Rose* oil (N = 201) | 1.06 | 0.91 | 0.14 | 1.00 | 2.00 | 2.00 | 2.00 | 4.97 |
| *Lavanda* oil (N = 255) | 0.68 | 0.76 | 0.14 | 0.28 | 1.00 | 2.00 | 2.00 | 4.00 |
| *Tea tree* oil (N = 150) | 0.73 | 0.95 | 0.14 | 0.28 | 1.00 | 2.45 | 2.50 | 4.49 |
| *Mentha* oil (N = 111) | 0.38 | 0.67 | 0.00 | 0.14 | 0.28 | 1.00 | 2.00 | 3.00 |
| *Lemon* oil (N = 133) | 0.32 | 0.46 | 0.06 | 0.14 | 0.35 | 1.00 | 1.00 | 2.66 |
| *Sandalwood* oil (N = 117) | 0.68 | 0.65 | 0.14 | 0.28 | 1.00 | 2.00 | 2.00 | 2.91 |
| *Frankincense* oil (N = 126) | 0.62 | 0.65 | 0.14 | 0.28 | 1.00 | 1.65 | 2.00 | 2.50 |
| *Ylang ylang* oil (N = 107) | 0.36 | 0.49 | 0.03 | 0.14 | 0.56 | 1.00 | 1.00 | 2.00 |
| *Eucalyptus* oil (N = 86) | 0.57 | 1.13 | 0.00 | 0.14 | 1.00 | 1.00 | 4.00 | - |
| *Jasmine* oil (N = 92) | 0.45 | 0.52 | 0.03 | 0.14 | 1.00 | 1.00 | 1.00 | - |

The differences in the frequency of use of all 11 types of EOs were analyzed by age groups. When no significant difference was observed, the samples were then pooled per the type of EO to determine the mean, standard deviation (SD), and selected percentiles (P25–P99). For example, among female consumers, the median frequency of use of Lavanda oil is 0.28, corresponding to twice a week. N is the number of female dermal users per EO. The Kruskall-Wallis test was conducted to compare difference in frequencies per age and per type of EO among females. When no statistical differences were observed, the sample was pooled per type of EO. For example, Ginger oil was excluded because of the difference observed in the frequency of use among age groups.

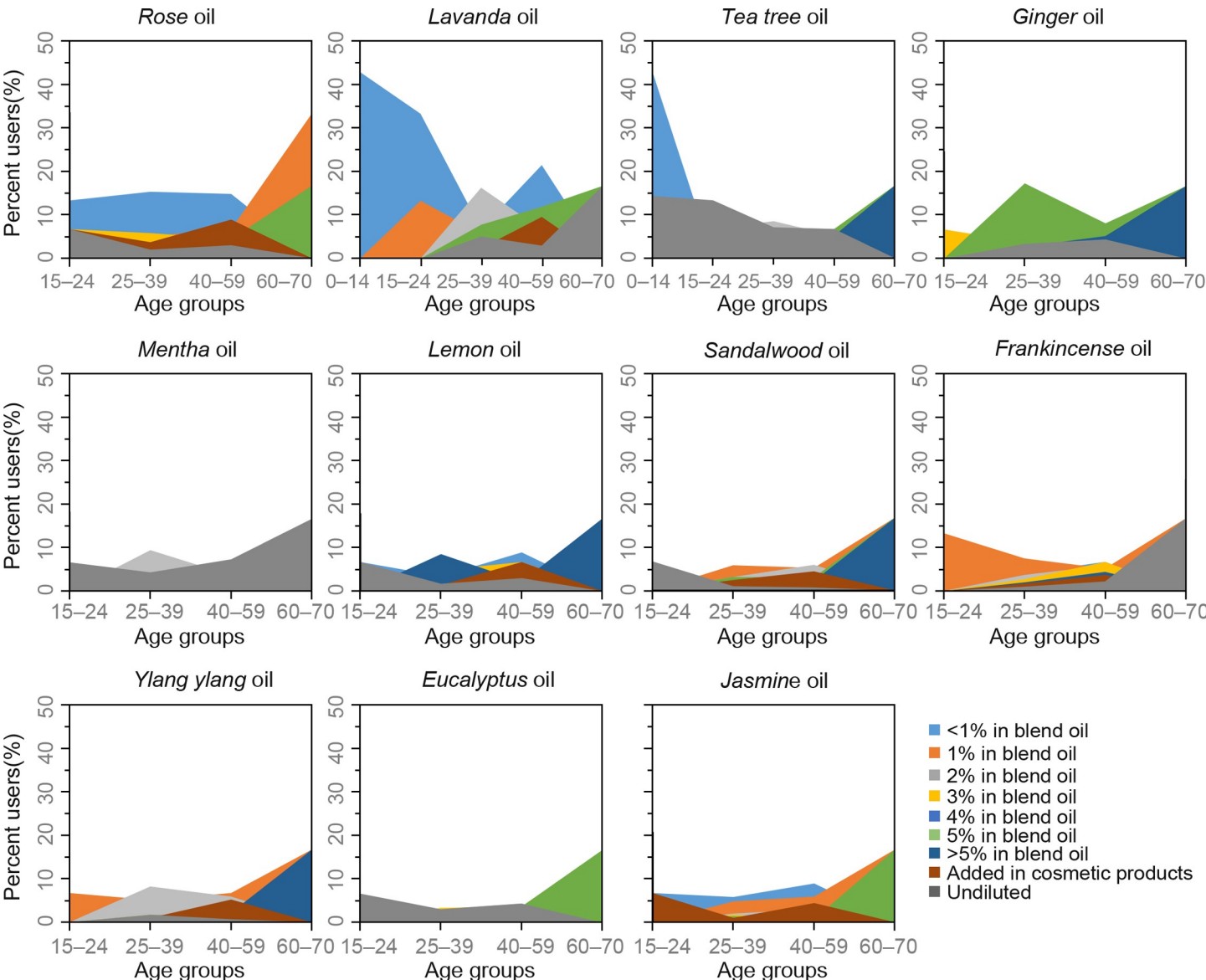

**Fig 3. Distribution of the prevalence of use and dilution rates of use in bases for the 11 types of essential oils across age groups.** For example, 9.5% of females diluted two drops of Mentha oil in 5 mL blend oil, which was at a 2% dilution rate. Notably, females aged 0–14 consumed Lavanda and Tea tree oil only.

used undiluted Tea tree oil, whereas females aged 60–70 used 5% and > 5% in blend oil (16.7%, 16.7%); 17.3% and 8.1% of females aged 25–39 and 40–59, respectively, used 5% of Ginger oil in blend oil, whereas females aged 60–70 used 5% and >5% in blend oil (16.7%, 16.7%). Moreover, for Mentha oil, except for females aged 25–39 (9.5%) who used EO at 2% dilution, other age groups tended to use undiluted ones. Finally, most users aged 15–59 used undiluted Eucalyptus, whereas users aged 60–70 used 5% of it in blend oil.

## Discussion

In this study, we provided a database on the usage patterns of 11 types of EOs. This database includes the general demographical data, prevalence of dermal use, percentages of users for the types of EOs used, exposed body areas, dilution rates of use, and frequency of use. To the

best of our knowledge, this comprehensive study is the first to provide data on EO usage patterns.

## Response rate

Generally, the response rates in web questionnaire surveys are lower than traditional ones [20]. The response rate of this study was 35.6%, which is higher compared with some European studies that also used web-based questionnaires [14, 21].

Different from the French study conducted on a representative panel of the general French population, our survey was conducted on aromatherapy consumers [18]. Although our results are not representative of the Chinese general population, the data obtained were valuable to generate the dermal exposure factor. Similar to the French study, a very low proportion of males (53 out of 534, approximately 9.9%) was observed, showing that men are less attracted to aromatherapy than women. In the French study, 60% (74 out of 123) of the female dermal user aged 25–59, whereas, in our study, the majority of female dermal users are aged 25–59 about 97% (443 out of 457, approximately 97%). The lower response in children can be explained by the lack of time of parents to answer the questionnaire, which could take 10 min.

## Usage patterns of aromatherapy EOs

The usage patterns of aromatherapy varied by age groups among female dermal users. For example, for the types of EO, a higher percentage of females aged 40–59 used Lavanda on the skin. Another example is the frequency use of Ginger oil. Females aged 60–70 used Ginger oil more frequently than other age groups; thus, Ginger oil was not included in Table 2 (S1 Fig).

Notably, consumers always practice their aromatherapy at home by diluting EO in bases (i.e., blend oil or cosmetics) for use, which is the major difference in usage pattern between formulated cosmetic products and aromatherapy products. Thus, our study included questions regarding the dilution rates of use, which is more accurate to determine the number of drops in bases. This case also makes our study different from the French study in 2016 [18]. Use of highly concentrated and undiluted EOs bears a risk of skin sensitization [22]. Due to lack of recommendations applied to aromatherapy EOs, females (15–70) tented to use undiluted Tea tree and Mentha oil. High percentages of females aged 60–70 used each EO at ≧5% dilution rates. This leaves further exposure and risk assessment to fragrance allergens in EOs highly necessary.

Through market promotion and recommendation from the aromatherapy websites, Lavanda oil has been known as one of the safest oils and helpful for skin disorders and anxiety [23, 24]. Thus, among the 11 types of EOs we studied, Lavanda oil was used by 55.8% of female dermal users as the most popular EO. In the French study, the highest percentage of females (60%) used Lavanda on skin [18]. That is also the reason why parents chose Lavanda oil for their children (0–14). Tea tree oil was another EO used by children. Contributing to their antiviral and antimicrobial activity, these two EOs were recommended to be applied on breast/ chest or back to relieve respiratory infection [25–27]. However, cases of contact dermatitis due to the tropical application of Tea tree oil or Lavanda oil or the combination of Tea tree and Lavanda oil were also reported [28–32]. Hence, further investigation is needed to determine which kind of fragrance allergens and the concentration of them to assess the skin allergy risk of Lavanda and Tea tree oils.

In this study, we report the application areas of 11 types of EOs. The results showed that higher percentages of females aged 25–59 used Rose, Lavanda, Sandalwood, Frankincense, and Jasmine oils on the whole face compared with other parts of body. This information is particularly important to assess sensitization risk because in use tests, the neck and face are more

sensitive than other parts of the body [5]. The face is a highly exposed body area, where the French study also described that 71% of adult females apply EOs on their face [18]. Research found that adult females aged 15–70, use on average nine different products on their face daily, with a 95th percentile exposed to 18 products [12]. The results in other studies indicated that gender (female) and age are risk factors for skin sensitization caused by fragrance allergens [33, 34]. Combining the mean values of use frequency per EO, we hypothesize that the majority of females aged 25–59 used Rose, Lavanda, Sandalwood, Frankincense, and Jasmine oil as kinds of facial care products (i.e., day/night cream) at home. Moreover, they tended to dilute EOs according to their preference which caused the diversity of dilution rates. In consideration of such usage pattern of EO, co-exposure to fragrance allergens contained in EOs and cosmetics could be one of the reasons why females above a certain age are more vulnerable to fragrance allergy.

## Limitations and uncertainties

We assumed that some consumers added EOs in cosmetic products, that is, night cream or body lotion. However, the question related to this topic was not specially formulated in the questionnaire. Moreover, the questionnaire referred to the general frequency and dilution rates of use for each EO, but the questionnaire should have been divided into specific parts of the body. For example, in one individual, Rose oil might be used on the face twice a day with 1% blend oil while it also being used on back once a week with 5% blend oil.

In addition, in our population sample, female consumers were overrepresented (N = 481; 90.1%) compared with males. Although we only used the data of female consumers to determine the factors for exposure assessment, females aged 0–14 and 60–70 were underrepresented. Therefore, some comparative analysis may be biased. A complementary study specific to children and the elderly would be necessary to define their exposure more precisely.

## Conclusion

This study describes the individual usage patterns of aromatherapy products among Chinese female consumers. In consideration of usage pattern of females aged 25–59, co-exposure to fragrance allergens contained in EOs and cosmetics makes them vulnerable to fragrance allergy. A database was created which can be helpful for the next step in calculating and assessing dermal exposure to fragrance allergens contained in EOs combined through dermal exposure modeling.

## Supporting information

**S1 Appendix. Questionnaire.** Usage patterns of aromatherapy among the Chinese population who are using essential oils.
(PDF)

**S1 Table. Demographical characteristics of the total consumers.**
(PDF)

**S1 Fig. Use frequency of Ginger oil across age groups.**
(PDF)

## Acknowledgments

The authors appreciate the assistance of Prof. Li Guangwu with the advice of the questionnaire design and an agreement to conduct a survey in the Chinese Aromatic and Aromatherapy

WeChat groups. The authors also would like to thank Enago (www.enago.jp) for the English language review.

## Author Contributions

**Supervision:** Satoshi Nakai.

**Writing – original draft:** Jun Xiao.

**Writing – review & editing:** Satoshi Nakai.

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
