## [Decision Letter · Decision Letter 0]

23 Feb 2022

PONE-D-21-39234Usage Patterns of Aromatherapy Essential Oil Among Chinese ConsumersPLOS ONE

Dear Dr. Nakai,

Thank you for submitting your manuscript to PLOS ONE. After careful consideration, we feel that it has merit but does not fully meet PLOS ONE’s publication criteria as it currently stands. Therefore, we invite you to submit a revised version of the manuscript that addresses the points raised during the review process.

We look forward to receiving your revised manuscript.

Kind regards,

Vineet Kumar Rai, PhD

Academic Editor

PLOS ONE

Journal Requirements:

3. Please amend your current ethics statement to address the following concerns:

a) Did participants provide their written or verbal informed consent to participate in this study?

b) If consent was verbal, please explain i) why written consent was not obtained, ii) how you documented participant consent, and iii) whether the ethics committees/IRB approved this consent procedure

“No authors have competing interest.”

Reviewers' comments:

Reviewer's Responses to Questions

**Comments to the Author**

1. Is the manuscript technically sound, and do the data support the conclusions?

Reviewer #1: Yes

Reviewer #2: Yes

2. Has the statistical analysis been performed appropriately and rigorously? 

Reviewer #1: No

Reviewer #2: Yes

3. Have the authors made all data underlying the findings in their manuscript fully available?

Reviewer #1: No

Reviewer #2: Yes

4. Is the manuscript presented in an intelligible fashion and written in standard English?

Reviewer #1: Yes

Reviewer #2: Yes

5. Review Comments to the Author

Reviewer #1: Current manuscript give a good account about the usage pattern of aromatherapy essential oil among Chinese consumers. Table 1, present the number and percentage of users through dermal routs per age groups. if possible to add in the first row of the table Frequency: n(%) instead of repeating the letter N in every row. also, Chi-square should be applied between different age group, at each gender, to evaluate the significance either within or between groups. an overall comment, table 1 should be re-disgned or reformatted to better express your results.

table 2, please add the significance and test used in the table

materials and methods doesn't include details about the data, either parametric or non parametric, what statistical analysis was used, to evaluate or assess what. also, sample size should presented in the materials and methods part, how many samples (participants) how sample size was calculated and a reference should be included

Reviewer #2: The study presents the results of original research. The article needs some revision to be suitable for publication. The general grammar should be revised to enhance clarity and enriched with recent references (2016-2022)

6. PLOS authors have the option to publish the peer review history of their article (what does this mean?). If published, this will include your full peer review and any attached files.

Reviewer #1: No

Reviewer #2: **Yes: **Benjamin Kingsley Harley

---

## [Author Response · Author response to Decision Letter 0]

27 Jun 2022

Responds to the reviewer’s comments:

Reviewer #1:

1. Comment: Current manuscript give a good account about the usage pattern of aromatherapy essential oil among Chinese consumers. Table 1, present the number and percentage of users through dermal routs per age groups. if possible to add in the first row of the table Frequency: n(%) instead of repeating the letter N in every row. also, Chi-square should be applied between different age group, at each gender, to evaluate the significance either within or between groups. an overall comment, table 1 should be re-disgned or reformatted to better express your results. table 2, please add the significance and test used in the table.

Response: It is really true as Reviewer suggested that Table 1 should be re-designed. We re-designed Table 1 in Line 185. 

It is reasonable to use Chi-square test to compare the differences between different age groups, at each gender. And we tried to use Chi-square test to evaluate the significance between or within different age groups at each gender, however, since the female dermal users were overrepresented (N = 457; 92.5% of total dermal users), and even within female dermal users, due to small sample size of female aged 0–14 and 60–70, the results of evaluation seems meaningless. Thus, the results are not shown in this manuscript.

Considering Reviewer’s suggestion on adding the significance and test used in the table, we added the Kruskall-Wallis test in the Table footnotes belong to Table 2 in Line 257-259.

2. Comment: materials and methods doesn't include details about the data, either parametric or non parametric, what statistical analysis was used, to evaluate or assess what.

Response: We are very sorry for our negligence of giving details on the data, as well as statistical analysis used. Thus, in revised manuscript, we have made correction according to the Reviewer’s comments.

Line 141-143, “Data on the type of EOs, exposed body areas, dilution rates of use are treated as non-parametric. And data on frequency of use for each EO is treated as parametric. ” were added.

Line 143-145, the statements of “The chi-squared or Fisher’s test was conducted to compare the differences in percentages of users per age groups.” were corrected as “The chi-squared or Fisher’s test was conducted to compare the differences in percentages of users on the type of EOs, exposed body area, and dilution rates of use across age groups. ”

Line 153-155, the statements of “The Kruskall-Wallis test was conducted to compare frequencies per age and per type of EO among females.” were corrected as “The Kruskall-Wallis test was conducted to compare difference in frequencies of use per age and per type of EO among females.” 

3. Comment: also, sample size should presented in the materials and methods part, how many samples (participants) how sample size was calculated and a reference should be included?

Response: It is really true as Reviewer suggested that we should explain the sample size in the material and methods parts, thus we corrected it as following:

Line 79, “ there are approximately 1,500 EO consumers registered to share” were added to explain the potential sample size. 

Special thanks to you for your good comments.

Reviewer #2: 

1. Comment: The study presents the results of original research. The article needs some revision to be suitable for publication. The general grammar should be revised to enhance clarity and enriched with recent references (2016-2022).

Response: As Reviewer suggested, we revised manuscript to fully meet PLOS ONE’s publication criteria, and we also added three recent references which were published in 2019, 2018, and 2021, respectively in Line 45: [8, 9] and Line 343: [32].

---

## [Editor Report · Decision Letter 1]

13 Jul 2022

Usage Patterns of Aromatherapy Essential Oil Among Chinese Consumers

PONE-D-21-39234R1

Dear Dr. Nakai,

We’re pleased to inform you that your manuscript has been judged scientifically suitable for publication and will be formally accepted for publication once it meets all outstanding technical requirements.

Kind regards,

Vineet Kumar Rai, PhD

Academic Editor

PLOS ONE
---

## [Editor Report · Acceptance letter]

3 Aug 2022

PONE-D-21-39234R1 

Usage Patterns of Aromatherapy Essential Oil Among Chinese Consumers 

Dear Dr. Nakai:

I'm pleased to inform you that your manuscript has been deemed suitable for publication in PLOS ONE. Congratulations! Your manuscript is now with our production department. 

Kind regards, 

on behalf of

Dr. Vineet Kumar Rai 

Academic Editor

PLOS ONE